# Zanubrutinib Ameliorates Cardiac Fibrosis and Inflammation Induced by Chronic Sympathetic Activation

**DOI:** 10.3390/molecules28166035

**Published:** 2023-08-12

**Authors:** Wenqi Li, Shuwen Zhu, Jing Liu, Zhigang Liu, Honggang Zhou, Qianyi Zhang, Yue Yang, Li Chen, Xiaowei Guo, Tiantian Zhang, Lingxin Meng, Dan Chai, Guodong Tang, Xiaohe Li, Cheng Yang

**Affiliations:** 1State Key Laboratory of Medicinal Chemical Biology, College of Pharmacy and Tianjin Key Laboratory of Molecular Drug Research, Nankai University, Haihe Education Park, 38 Tongyan Road, Tianjin 300353, China; 1120210653@mail.nankai.edu.cn (W.L.); 2120211341@mail.nankai.edu.cn (S.Z.); 2120221626@mail.nankai.edu.cn (J.L.); 1120220741@mail.nankai.edu.cn (Z.L.); honggang.zhou@nankai.edu.cn (H.Z.); zqy15027605203@163.com (Q.Z.); 15502617145@163.com (Y.Y.); chenliesther0620@163.com (L.C.); gxw990829@163.com (X.G.); zhangtt_gala@163.com (T.Z.); 13356720390@163.com (L.M.); C2094760933@163.com (D.C.); 2Tianjin International Joint Academy of Biomedicine, Tianjin 300457, China; 3Department of Cardiology, Beijing Hospital, National Center of Gerontology, Institute of Geriatric Medicine, Chinese Academy of Medical Sciences, Beijing 100730, China

**Keywords:** heart failure, β-adrenergic receptor, cardiac fibrosis, cardiac inflammation, Bruton’s tyrosine kinase, Zanubrutinib

## Abstract

(1) Background: Heart failure (HF) is the final stage of multiple cardiac diseases, which have now become a severe public health problem worldwide. β-Adrenergic receptor (β-AR) overactivation is a major pathological factor associated with multiple cardiac diseases and mediates cardiac fibrosis and inflammation. Previous research has demonstrated that Bruton’s tyrosine kinase (BTK) mediated cardiac fibrosis by TGF-β related signal pathways, indicating that BTK was a potential drug target for cardiac fibrosis. Zanubrutinib, a second-generation BTK inhibitor, has shown anti-fibrosis effects in previous research. However, it is unclear whether Zanubrutinib can alleviate cardiac fibrosis induced by β-AR overactivation; (2) Methods: In vivo: Male C57BL/6J mice were treated with or without the β-AR agonist isoproterenol (ISO) to establish a cardiac fibrosis animal model; (3) Results: In vivo: Results showed that the BTK inhibitor Zanubrutinib (ZB) had a great effect on cardiac fibrosis and inflammation induced by β-AR. In vitro: Results showed that ZB alleviated β-AR-induced cardiac fibroblast activation and macrophage pro-inflammatory cytokine production. Further mechanism studies demonstrated that ZB inhibited β-AR-induced cardiac fibrosis and inflammation by the BTK, STAT3, NF-κB, and PI3K/Akt signal pathways both in vivo and in vitro; (4) Conclusions: our research provides evidence that ZB ameliorates β-AR-induced cardiac fibrosis and inflammation.

## 1. Introduction

Heart failure is a severe disease that has a major clinical and economic impact on the world’s population [1,2]. The overactivation of the sympathetic nervous system is the main pathological factor in heart failure and plays a key role in the occurrence and development of heart failure, including cardiac fibrosis and cardiac inflammation, and other pathological manifestations [3,4].

β-adrenergic receptor (β-AR), a classical G-protein-coupled receptor, is also an important neuroendocrine receptor and a major effector of cardiac sympathetic stress, which plays an essential role in maintaining the physiological function of the heart [5]. During heart failure, continuous activation of β-AR causes a continuous increase in cellular Ca^2+^ concentration, leading to the formation of calcium/calmodulin-dependent protein kinase (CaMK II). The β-AR-Gs-CaMK II signaling pathway is activated, thereby promoting cardiac hypertrophy, apoptosis, and other myocardial remodeling processes, and finally causing heart failure [6]. At the same time, β-AR can also activate the phosphatidylinositol 3-kinase (PI3K)/protein kinase B (Akt) cell survival pathway by activating Gi proteins, which have protective effects on the heart [7]. Isoproterenol (ISO), a β-AR agonist, has been widely used in the establishment of cardiac fibrosis models [8]. Clinically, β-blocker, as one of the golden triangle drugs in heart failure treatment, has shown promising results in the treatment of patients with heart failure. However, it still has some limitations in clinical application, such as serious side effects [9,10]. Therefore, how to maintain the physiological function of the heart while improving the pathological effect has become an urgent problem for the treatment of heart failure in the future.

Bruton’s tyrosine kinase (BTK) belongs to the family of non-receptor tyrosine kinases (nRTKs) in hepatocellular carcinoma and was initially recognized to play a key role in the development and function of B cells [11,12]. BTK has been reported to be associated with a variety of severe human diseases, including chronic lymphocytic leukemia and certain hyperactivated inflammatory responses after infection [13,14]. Interestingly, previous studies have shown that some members of the TEC kinase family are involved in the development and progression of cardiovascular diseases. The bone marrow tyrosine kinase gene (BMX) is involved in the occurrence of cardiac hypertrophy under pressure overload [15,16]. In addition, inhibition of IL-2-induced T cell kinase (ITK) can reduce Th1/Th17 response after heart transplantation and significantly prolong the mean survival time of transplanted hearts [17]. However, the role of BTK in non-immune cells and other diseases remains unclear. The latest research shows that BTK might be a potential drug target for cardiac fibrosis [18]. Zanubrutinib (ZB), a second-generation BTK inhibitor, is currently used in the treatment of capsid cell lymphoma (MCL) and chronic lymphocytic leukemia/small lymphocytic lymphoma (CLL/SLL) [19]. ZB is highly selective for BTK, a property that is intended to increase drug potency and reduce off-target effects [20,21].

Therefore, our research focuses on determining whether ZB has a therapeutic effect on ISO-induced cardiac fibrosis and inflammation both in vivo and in vitro. Then, we further explored the mechanism of ZB in cardiac fibrosis and inflammation both in vivo and in vitro.

## 2. Results

### 2.1. ZB Alleviated ISO-Induced Cardiac Dysfunction In Vivo

Overactivation of the sympatho-adrenergic system is a major pathological factor in cardiac diseases. Cardiac β-AR overactivation, which is central to the development of cardiac diseases, can cause cardiac fibrosis and cardiac inflammation [4,22]. Interestingly, the upregulated BTK is mainly localized in cardiac fibroblasts in response to hypertensive or ischemic stimulation, indicating that BTK might be a potential therapeutic target for cardiac fibrosis [19]. Therefore, ZB, a second-generation BTK inhibitor, was used to verify the biological function of ISO-induced cardiac fibrosis and inflammation. The cardiac contractile function is a vital part of cardiac function. The ejection fraction (EF%) and fractional shortening were calculated based on the parameters measured from M-mode images to reflect the systolic cardiac function. To determine whether ZB alleviates cardiac dysfunction caused by sympathetic stress, cardiac systolic function was measured in mice. As expected, ISO induced structural and functional cardiac changes (Figure 1b). In addition, we found that left ventricular ejection fraction (LVEF%) and left ventricular fractional shortening (LVFS%) were decreased and diastolic left ventricular posterior wall thickness (LVPW;d) was increased. These data suggested that ISO induced cardiac systolic dysfunction, while treatment with ZB (20 mg/kg) reversed this process (Figure 1c–e). In addition, the cardiomyocyte hypertrophy indicators, including ANP and BNP, were increased in ISO-treated mice, while decreased by treatment with ZB (20 mg/kg) (Figure 1f,g). Overall, these data indicate that ZB alleviates ISO-induced cardiac fibrosis in vivo.

### 2.2. ZB Alleviated ISO-Induced Cardiac Fibrosis In Vivo

Cardiac fibroblasts are the main effector cells of cardiac fibrosis. Our results further confirmed that phospho-BTK increased in ISO-treated CFs, indicating BTK might play an essential role in β-AR-induced cardiac fibrosis (Appendix A). Therefore, we explored whether a small molecule inhibitor of BTK had an effect on β-AR-induced cardiac fibrosis. Firstly, we observed general indicators of cardiac fibrosis, including heart size, the ratio of heart weight to body weight (HW/BW), and the ratio of heart weight to tibia length (HW/TL). As expected, ISO markedly increased heart size, while treatment with ZB (20 mg/kg) reversed this process (Figure 2b). Similarly, ZB (20 mg/kg) attenuated HW/BW and HW/TL (Figure 2c,d), which were both increased significantly by ISO. Furthermore, we detected the pathological index of cardiac fibrosis. As revealed by histological analysis of picrosirius red, the area of cardiac fibrosis was increased in mouse hearts in the ISO group, while inhibited by treatment with ZB (20 mg/kg) (Figure 2e). In addition, the activation of cardiac fibroblasts indicators, including connective tissue growth factor (CTGF), Fibronectin (Fn), α-SMA, Collagen-I (Col-I), and Collagen-III (Col-III), were increased in ISO-treated mice, while decreased by treatment with ZB (20 mg/kg) (Figure 2f,j). Overall, these data indicate that ZB alleviates ISO-induced cardiac fibrosis in vivo.

### 2.3. ZB Reduced ISO-Induced Cardiac Fibroblasts Activation In Vitro

As mentioned above, ZB inhibited ISO-induced cardiac fibrosis in vivo. Since the activation of cardiac fibroblasts (CFs) is a key factor in cardiac fibrosis, which includes CFs proliferation, trans-differentiation, and extracellular matrix (ECM) synthesis [23], we tested the regulatory effects of ZB on β-AR-dependent activation of CFs. As expected, CF proliferation levels, including cell viability and mRNA level of CTGF, were increased with ISO treatment (10 μmol L^−1^), while decreased after pretreatment with ZB (1 μmol L^−1^) in NRCFs (Figure 3d,e). Furthermore, CF trans-differentiation levels, including Fn and α-SMA, were increased with ISO treatment (10 μmol L^−1^), while decreased pretreatment with ZB (1 μmol L^−1^) in NRCFs (Figure 3f–h). In addition, the ECM synthesis levels of CFs, including mRNA levels of Col-I and Col-III, demonstrated similar results (Figure 3i,j). Together, these data indicate that ZB reduces ISO-induced cardiac fibrosis in vitro.

### 2.4. ZB Alleviated ISO-Induced Cardiac Inflammation In Vivo

Cardiac fibrosis is commonly accompanied by cardiac inflammation [24]. Thus, we next investigated whether ZB could alleviate ISO-induced cardiac inflammation in vivo. Macrophages, which can secrete inflammatory cytokines, are the main effector cells of cardiac inflammation [8]. As revealed by immunohistochemical staining of the macrophage marker Mac-3, the Mac-3-positive area in the heart was increased in the ISO group, while inhibited by treatment with ZB (20 mg/kg) (Figure 4b). Consistently, ZB treatment (20 mg/kg) also reduced the ISO-induced increase in pro-inflammatory cytokines IL-1β, IL-6, and TNF-α both in terms of mRNA and protein levels (Figure 4c–h). Overall, these data indicate that ZB alleviates ISO-induced cardiac inflammation in vivo.

### 2.5. ZB Reduced ISO-Induced Macrophage Pro-Inflammatory Production In Vitro

As mentioned above, ZB inhibited ISO-induced inflammation in vivo. Thus, we next investigated whether ZB had an effect on β-AR-induced cardiac inflammation. Since the secretion of pro-inflammatory cytokines is an essential part of cardiac inflammation during the overactivation of the sympatho-adrenergic system [25], we investigated pro-inflammatory cytokines in β-AR-induced cardiac inflammation. As expected, mRNA levels of pro-inflammatory cytokines IL-1β, IL-6, and TNF-α were increased with ISO treatment (10 μmol L^−1^), while decreased with pretreatment with ZB (1 μmol L^−1^) in RAW264.7 (Figure 5c–e). Consistently, ELISA of cell supernatants demonstrated the same results (Figure 5f–h). Together, these data indicate that ZB reduces ISO-induced inflammation in vitro.

### 2.6. ZB Reduced ISO-Induced Cardiac Fibroblasts Activated by STAT3 and PI3K/Akt Signaling Pathways

To clarify the underlying mechanism of the anti-fibrosis effect of ZB in vitro, some typical signal pathways in β-AR-induced cardiac fibroblasts activation, including STAT3 and PI3K/Akt [26,27], were examined. As expected, ISO remarkably increased BTK phosphorylation levels, while ZB, a classical BTK inhibitor, reduced this process (Figure 6b,c). In terms of indicators of CF proliferation, Western Blot showed a remarkable increase in STAT3 phosphorylation treatment with ISO (10 μmol L^−1^), while a significant decrease with pretreatment with ZB (1 μmol L^−1^) (Figure 6b,d), suggesting that ZB reduced ISO-induced cardiac fibroblast proliferation by the STAT3 signaling pathway. In terms of indicators of CF trans-differentiation and ECM synthesis, Western Blot showed a remarkable increase in PI3K and Akt phosphorylation treatment with ISO (10 μmol L^−1^), while a significant decrease with pretreatment with ZB (1 μmol L^−1^) (Figure 6b,e,f), suggesting that ZB reduced ISO-induced CF trans-differentiation and ECM synthesis by the PI3K/Akt signaling pathways. Together, these data indicate that ZB reduces ISO-induced CF activation by the STAT3 and PI3K/Akt signaling pathways.

### 2.7. ZB Reduced ISO-Induced Macrophage Pro-Inflammatory Production by PI3K/Akt and NF-κB Signaling Pathways

To clarify the underlying mechanism of the anti-inflammatory effect of ZB in vitro, some typical signal pathways in β-AR-induced macrophage inflammation, including PI3K/Akt and NF-κB [28,29], were examined. As expected, ISO remarkably increased BTK phosphorylation levels, while ZB reduced this process (Figure 7b,c). In terms of indicators of macrophage pro-inflammatory IL-1β production, Western Blot showed a remarkable increase in PI3K and Akt phosphorylation treatment with ISO (10 μmol L^−1^), while a significant decrease with pretreatment with ZB (1 μmol L^−1^) (Figure 7b,d,e), suggesting that ZB reduced ISO-induced macrophage pro-inflammatory IL-1β production by the PI3K/Akt signaling pathway. In terms of indicators of macrophage pro-inflammatory IL-6 and TNF-α, Western Blot showed a remarkable increase in NF-κB phosphorylation treatment with ISO (10 μmol L^−1^), while a significant decrease with pretreatment with ZB (1 μmol L^−1^) (Figure 7b,f), suggesting that ZB reduced ISO-induced macrophage pro-inflammatory IL-6 and TNF-α production by the NF-κB signaling pathway. Overall, these data indicate that ZB reduces ISO-induced macrophage pro-inflammatory production by the PI3K/Akt and NF-κB signaling pathways.

### 2.8. ZB Reduced ISO-Induced Cardiac Fibrosis and Inflammation by STAT3, NF-κB, and PI3K/Akt Signaling Pathways In Vivo

To clarify whether ZB has potential anti-fibrosis and anti-inflammatory effects via the BTK, STAT3, NF-κB, and PI3K/Akt pathways in vivo, we explored these signaling pathways in heart tissue. Similarly, ISO remarkably increased BTK phosphorylation levels in mice heart tissue, while ZB reduced this process (Figure 8b,c). ISO also increased STAT3, NF-κB, PI3K, and Akt phosphorylation levels, while ZB reduced this process (Figure 8d–g). Overall, these data indicate that ZB reduces ISO-induced cardiac fibrosis and inflammation by the BTK, STAT3, NF-κB, and PI3K/Akt signaling pathways in vivo.

### 2.9. Working Model of the Effects of ZB on β-AR Stimulation-Induced Cardiac Fibrosis and Inflammation

In summary, our study demonstrates that ZB inhibits ISO-induced fibrosis and inflammation both in vitro and in vivo by regulating the STAT3, NF-κB, and PI3K/Akt signaling pathways by targeting BTK (Figure 9). These findings reinforce that ZB could be a candidate compound for the development of potential anti-cardiac fibrosis and anti-cardiac inflammation drugs to improve the medical treatment of cardiac fibrosis and inflammation.

## 3. Discussion

BTK is a non-receptor cytoplasmic tyrosine kinase mainly expressed in pre-B cells and B-lymphocytes [12]. Previous study has shown that the upregulated BTK is primarily localized in cardiac fibroblasts in response to hypertensive or ischemic stimulation. Loss of the expression or pharmacological inhibition of BTK alleviates the process of myocardial fibrosis and cardiac dysfunction under various pathological conditions, suggesting the relationship between the amount of BTK and myocardial fibrosis [19]. As a tyrosine kinase, BTK functions through phosphorylation. However, the role of phosphorylated BTK in myocardial fibrosis deserves to be explored. Moreover, CFs are the main effector cells of cardiac fibrosis, with the activation of CFs being a key factor in cardiac fibrosis. Additionally, as a widely distributed receptor in the heart, β-AR regulates a variety of CF behaviors, including proliferation, trans-differentiation, and ECM synthesis. By affecting these functions, β-AR promotes cardiovascular diseases and enhances cardiovascular cell growth and metastasis in vivo [30,31]. Interestingly, the phosphorylation of BTK increased in β-AR-induced cardiac fibroblasts, which indicated the relationship between the activity of BTK and cardiac fibrosis. ZB is a second-generation small molecule inhibitor of BTK and can form a covalent bond with the cysteine residue in the active site of BTK, thereby inhibiting BTK activity [32]. Our research demonstrated that ZB inhibited BTK phosphorylation in a cardiac fibrosis model both in vivo and in vitro. We further explored whether blocking BTK alleviated cardiac fibrosis caused by β-AR.

As expected, ZB alleviated β-AR-induced cardiac fibrosis by the STAT3 and PI3K/Akt signal pathways both in vivo and in vitro. So far, little is known about the role of BTK in non-immune cells and the relationship between BTK and β-AR-induced myocardial fibrosis. Meanwhile, recent studies have shown that BTK directly binds to phosphorylated TGF-β receptor I (TβRI) at tyrosine 182 in cardiac fibrosis due to pressure overload, and thereby modulates the activation of downstream SMAD-dependent or SMAD-independent TGF-β signaling, leading to enhanced myofibroblast transition and ECM gene overexpression [19]. In brief, BTK might be a potential drug target for cardiac fibrosis. In addition, previous studies indicate that ibrutinib, a first-generation small molecule inhibitor of BTK, has an anti-fibrotic effect in multiple organ fibrosis diseases, containing tumor fibrosis [33], liver fibrosis [34], and chronic pancreatitis [35]. Meanwhile, the STAT3 and PI3K/Akt signaling pathways are two of the most classical downstream signaling pathways of β-AR that regulate cardiac fibrosis. As a second-generation inhibitor of BTK, our study demonstrated that ZB regulated cardiac fibrosis by BTK-mediated STAT3 and PI3K/Akt signal pathways both in vivo and in vitro. In a cardiac fibrosis model caused by chronic sympathetic overactivation, whether ZB alleviates cardiac fibrosis through the TGF-β signaling pathway needs to be further studied. Moreover, since BTK is an essential molecule of the B-cell receptor signaling pathway, BTK inhibitors may also exert anti-fibrotic effects by reducing B-cell-related inflammation or other intrinsic events in cardiac fibrosis under pathological conditions [28]. Finally, ZB may also affect ISO-induced cardiac fibrosis in mice by modulating B-cell activation, which needs further research and verification.

Cardiac fibrosis is commonly accompanied by cardiac inflammation [24]. The NF-κB and PI3K/Akt signaling pathways are two of the most classical downstream signaling pathways of β-AR that regulate cardiac inflammation. Thus, we further investigated whether blocking BTK alleviated cardiac inflammation caused by β-AR. As expected, ZB alleviated β-AR-induced cardiac inflammation by the NF-κB and PI3K/Akt signal pathways both in vivo and in vitro. Previous studies indicate that ibrutinib, a first-generation small molecule inhibitor of BTK, ameliorates pulmonary fibrosis by inhibiting inflammation [36]. Other research demonstrates that ibrutinib reduces the activation of NF-κB and NLRP3 inflammasomes by targeting BTK and finally ameliorates pulmonary inflammation [37], suggesting that BTK might be a potential drug target for anti-inflammation. However, ibrutinib may have an off-target effect in a mice pulmonary fibrosis model, which could not represent the role of BTK in organ fibrosis. ZB, which demonstrates a better on-target effect on BTK compared to ibrutinib, was shown to reduce cardiac inflammation both in vivo and in vitro. Moreover, since BTK is an essential molecule of the B-cell receptor signaling pathway, BTK inhibitors may also exert an anti-inflammatory effect by reducing B-cell-related inflammation. Finally, ZB may also affect ISO-induced cardiac inflammation in mice by modulating B-cell activation, which needs further research and verification.

Heart failure is the final stage of many heart diseases, and cardiac remodeling is a necessary pathological process in heart failure [38]. ISO, as a β-AR agonist, is able to mimic cardiac remodeling caused by continuous activation of the sympathetic nervous system. Previous studies have shown that ISO constructs a classical animal model of sustained activation of the sympathetic system, which mainly displays cardiac inflammation on 0–3 days, cardiac fibrosis on 3–7 days, and heart failure at about 1 month [22,24]. In our study, our animal model induced with sustained ISO for 7 days mimicked the early stages of cardiac remodeling and may not have reached the pathological process of heart failure. In addition, how ZB regulates β-AR via BTK is unknown, which needs further study.

## 4. Materials and Methods

### 4.1. Animal Model and Treatments

Male C57BL/6J mice (8 to 12 weeks old; 20–25 g body weight) and Male Sprague Dawley neonatal rats (1 to 3 days old) were purchased from Weitonglihua Company (Beijing, China). All animal care and experimental procedures were approved by the Institutional Animal Care and Use Committee (IACUC) of Nankai University (Permit No. SYXK 2014-0003). Mice were fed in a 12 h light/dark cycle in a room with controlled temperature (22–26 °C) and humidity (60 ± 2%) and had open access to food and water.

In animal experiments, 30 mice were randomly divided into 5 groups (Appendix A): Saline group, ISO (Sigma-Aldrich, I5627, St. Louis, MO, USA)group (10 mg kg^−1^ d^−1^), Metoprolol (Met) group (ISO + Met) (30 mg kg^−1^ d^−1^), Zanubrutinib (ZB) 10 mg/kg group (ISO + ZB 10 mg/kg) (10 mg kg^−1^ d^−1^), and ZB 20 mg/kg group (ISO + ZB 20 mg/kg) (20 mg kg^−1^ d^−1^). Met (MedChemExpress, HY-17503, Shanghai, China) was used as the positive control. A cardiac fibrosis model was established by daily subcutaneous injection of 10 mg/kg isoproterenol (ISO) (Sigma-Aldrich, I5627, USA) for 7 days [22]. For the ISO group, 10 mg/kg ISO (Sigma-Aldrich, USA) was injected subcutaneously for 7 days, while the Saline group was given an equal volume of Saline. Met (MedChemExpress, HY-17503, China) and ZB (MedChemExpress, HY-101474A, China) were dissolved in Saline, ultrasound shattered and then administered by gavage once a day for 7 days before daily ISO injection [22,24].

### 4.2. Echocardiographic Measurements

Echocardiography was used to evaluate cardiac function on the 8th day after ISO administration for 24 h. The Vevo 2100 system (VisualSonics Inc., Toronto, ON, Canada) was used to collect data. Mice were anesthetized with 1.5% isoflurane (Baxter Healthcare Corp., New Providence, RI, USA, R510-22) until the heart rate was at 400–500 beats per minute. The thickness and chamber dimensions were determined from M-mode images acquired at the mid-papillary level in the parasternal short-axis view and B-mode images acquired in the parasternal long- and short-axis views. The ejection fraction (EF%) and fractional shortening were used to represent the systolic cardiac function. The diastolic left ventricular posterior wall thickness (LVPW;d) was measured based on the parameters from M-mode, namely, the left ventricular ejection fraction (LVEF) and left ventricular fractional shortening (LVFS). All data were averaged from three consecutive cardiac cycles.

### 4.3. Isolation of Cardiac Fibroblasts

Primary neonatal rat cardiac fibroblasts (NRCFs) were isolated from 1- to 2-day-old Sprague Dawley rats obtained from Weitonglihua Company (Beijing, China). Neonatal rats were euthanized under the standard protocol. Harvested hearts were washed with sterile PBS and minced in ice-cold PBS. Minced heart tissues were digested with digestion buffer prepared with 0.06% (*w*/*v*) pancreatin (Gibco, 27250018, Waltham, MA, USA) and 0.05% (*w*/*v*) collagenase type II (Gibco, 17101015, Waltham, MA, USA) in PBS. Tissue digestion was performed by incubating minced hearts in digestion solution for 30 min at 37 °C. The collected cells were spun down and resuspended in culture medium: DMEM with 10% FBS (Gibco, 10100147, USA) and penicillin–streptomycin (Gibco, 15070063, USA). After washing out digestion buffer with culture medium twice, cells were seeded on cell culture dishes and incubated for 2 h in the incubator (37 °C, 5% CO_2_, 90% humidity). After 2 h incubation, floating cells and culture medium were removed and attached cells were continued to culture with fresh culture medium. NRCFs at passage 2 were used for this study.

### 4.4. Histological Analysis

Mice were anesthetized and sacrificed after echocardiography analysis. The hearts were excised and weighed immediately after being washed with cold PBS. The cardiac tissues for histological analysis were fixed with 4% paraformaldehyde for 12 h, dehydrated in 20% sucrose for 24 h, and then embedded in paraffin. Paraffin-embedded hearts were cut into 5 µm thick sections on a microtome and transferred onto glass slides. Then, serial sections (5 μm thick) were stained with hematoxylin and eosin stain (H&E) for morphological analysis, and picrosirius red (PSR) was used for the detection of fibrosis according to the manufacturer’s instructions [8]. The percentage of collagen area was measured using a quantitative digital image analysis system (Image-Pro Plus 6.0) to evaluate cardiac fibrosis.

### 4.5. Total RNA Extraction and Real-Time PCR Analysis

Total RNA was isolated from heart tissues using Trizol Reagent (Invitrogen, 10296028CN, Waltham, MA, USA), and then 1 µg RNA was reverse transcribed into cDNA using the Reverse SYBR Select Master Mix kit (Tiangen, KR118, Jiulong County, China) followed by fluorescence quantitative real-time PCR (Yeasen, 11198ES08, Shanghai, China). Relative mRNA levels were normalized to GAPDH and then quantified using the comparative 2^−ΔΔCT^ method, and fold changes were compared to the control. The primers used in this study are given in Table 1.

### 4.6. Cell Culture and Treatments

RAW264.7 was cultured in DMEM medium, supplemented with 10% FBS and 1% penicillin and streptomycin, and maintained at 37 °C with 5% CO_2_. NRCFs were cultured in DMEM medium, supplemented with 15% FBS and 1% penicillin and streptomycin, and maintained at 37 °C with 5% CO_2_. NRCFs were prepared from neonatal Sprague Dawley rats as previously described. Before being treated with ISO (10 μmol L^−1^) (dissolved in sterile water), RAW264.7 was starved for 12 h with serum-free medium, and NRCFs were starved for 24 h with serum-free medium. In the cases of Met (10 μmol L^−1^) and ZB (0.5 or 1 μmol L^−1^) (both dissolved in DMSO), cells were pretreated with Met or ZB for 1 h before ISO. The concentration of ZB was selected according to (Figure 2c and Figure 4b).

### 4.7. Western Blot Analysis

Cells and heart tissue were lysed in protein lysis (1% Deoxycholic acid, 10 mM Na_4_P_2_O_7_, 1% TritonX-100, 10% Glycerol, 100 mM NaCl, 5 mM EDTA (pH = 8.0), 20 mM Tris-HCl (pH = 7.4), 0.1% SDS, 50 mM NaF, 1 mM Na_3_VO_4_, 1 mM PMSF, 10 mg/L aprotinin). Equal amounts of protein were separated on 10% SDS-PAGE gels and transferred to nitrocellulose membranes. After blocking with 5% milk for 1 h, the membranes were incubated with primary antibodies at 4 °C overnight and incubated with secondary antibodies for 1 h at room temperature. After being treated with corresponding horseradish peroxidase (HRP)-conjugated secondary antibodies (Abacm, ab97051, Cambridge, MA, USA), the protein bands were detected using a chemiluminescence reagent (Affinity, KF8003, San Francisco, CA, USA) and visualized on a GeneGnome chemiluminescent imaging system (Syngene, Iselin, NJ, USA) [9]. The relative density of each band was analyzed using ImageJ. β-Tubulin and GAPDH served as internal references in densitometric analysis. The primary antibodies used in this study are given in Table 2.

### 4.8. Immunofluorescence

All the steps were the same as previously described. After pretreatment with Met (10 μmol L^−1^) or ZB (0.5 μmol L^−1^ and 1 μmol L^−1^) for 1 h, ISO (10 μmol L^−1^) was added in the culture medium for 5 min, 15 min, and 24 h. Then, the cells were fixed, and the nuclei were stained with DAPI (Solarbio, S2110, Beijing, China). Fluorescence was examined with a confocal microscope (Nikon, Tokyo, Japan).

### 4.9. Immunohistochemistry

Mouse hearts were collected on day 8 after ISO injection, harvested, washed with cold phosphate-buffered Saline, fixed with 4% paraformaldehyde for 6–8 h, and embedded in paraffin. Serial sections (5 μm thick) were stained with antibodies against the macrophage marker Mac-3 (1:200 dilution) (BD Biosciences, 550292, San Jose, CA, USA). Tissue sections were imaged using the NanoZoomer-SQ (Hamamatsu, Japan). Ten fields were randomly selected from the image of each section, and the areas of the Mac-3-positive area were measured with Image-Pro Plus 6.0. The ratio of positive-stained area to total myocardial area was calculated to determine the infiltration of macrophages. The UltraSensitiveTM SP (Mouse/Rabbit) IHC Kit (Maxim, KIT-97, Shanghai, China) and DAB Kit (Maxim, DAB-0031, Fuzhou, China) were used according to the instructions to detect the expression of Mac-3 [20].

### 4.10. Enzyme-Linked Immunosorbent Assay (ELISA)

The protein levels of Interleukin-1β (IL-1β) (Jianglai Bio, JL18442, Shanghai, China), Interleukin-6 (IL-6) (Jianglai Bio, JL20268, Shanghai, China) and Tumor necrosis factor (TNF-α) (Jianglai Bio, JL10484, Shanghai, China) in heart tissue and the supernatant of RAW264.7 were measured using ELISA kits (Jianglai Bio, JL18442, Shanghai, China). Procedures were conducted according to the manufacturer’s instructions, and absorbances were read at 450 nm (Microplate Reader Model 550, Bio-Rad, Hercules, CA, USA). All concentration values were in the linear range of the standard curve, and the final results are shown as the ratio of total protein concentrations.

### 4.11. Cell Counting Kit-8 Assay

The cell viability following ISO treatment was measured with Cell Counting Kit-8 (CCK-8) (Dojindo, CK04, Tokyo, Japan). The operations were conducted according to kit instructions. Cardiac fibroblasts of 5000 per well were cultured in 96-well plates overnight. CCK-8 solutions were added and incubated for 3 h. The OD450 nm was detected in a Microplate Reader Model 550 (Bio-Rad, Hercules, CA, USA), and the cell viability and inhibition of cardiac fibroblasts were calculated.

### 4.12. Statistical Analysis

Data were expressed as mean ± SEM. Differences between more than two groups were analyzed by one-way analysis of variance (ANOVA) with Tukey’s post-hoc multiple comparison tests. *p* < 0.05 was considered statistical significance.

## 5. Conclusions

In summary, our study demonstrates that ZB inhibits ISO-induced fibrosis and inflammation by regulating the STAT3, NF-κB, and PI3K/Akt signaling pathways via targeting BTK both in vitro and in vivo (Figure 9). These findings reinforce that ZB could be a candidate compound for the development of potential anti-cardiac fibrosis and anti-cardiac inflammation drugs to improve the medical treatment of cardiac fibrosis and inflammation.

## Figures and Tables

**Figure 1 molecules-28-06035-f001:**
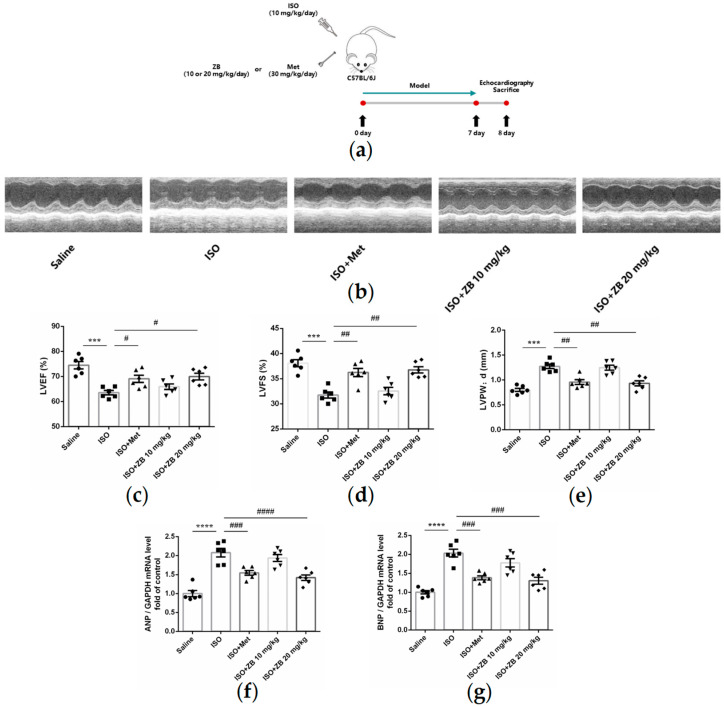
ZB prevents ISO-induced cardiac dysfunction. (**a**) Dosing regimen in ISO-induced cardiac fibrosis model. Mice were treated by daily subcutaneous injection of 10 mg kg^−1^ d^−1^ ISO for 7 days. For 7 days, 10 mg/kg, 20 mg/kg ZB, 30 mg/kg Met or equal volumes of Saline was gavaged daily as indicated. Met was used as a positive control; (**b**) Representative echocardiographic M-mode images of left ventricles from mice at day 8; (**c**) Echocardiographic measurement of LVEF (*n* = 6); (**d**) Echocardiographic measurement of LVFS (*n* = 6); (**e**) Quantitative analysis of LVPW;d (*n* = 6); (**f**) The mRNA levels of ANP in heart tissues (*n* = 6); (**g**) The mRNA levels of BNP in heart tissues (*n* = 6); Quantification of ANP and BNP were normalized to GAPDH. The data are shown as mean ± SEM (one-way ANOVA with Tukey’s post-hoc multiple comparison tests). ***, *p* < 0.001, ****, *p* < 0.0001 vs. Saline; #, *p* < 0.05, ##, *p* < 0.01, ###, *p* < 0.001, ####, *p* < 0.0001 vs. ISO. ISO, isoproterenol; Met, metoprolol; ZB, Zanubrutinib; LVEF, left ventricular ejection fraction; LVFS, left ventricular fractional shortening; LVPW;d, diastolic left ventricular posterior wall thickness.

**Figure 2 molecules-28-06035-f002:**
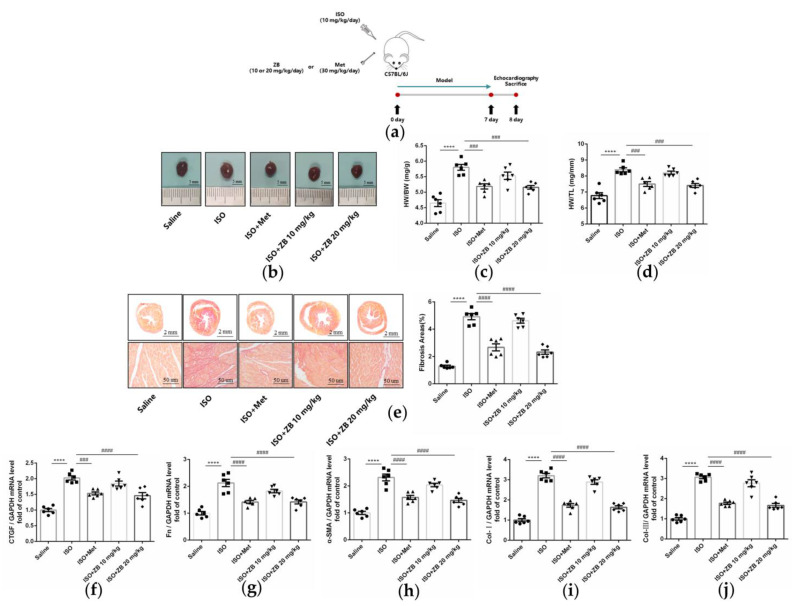
ZB prevents ISO-induced cardiac dysfunction. (**a**) Dosing regimen in ISO-induced cardiac fibrosis model. Mice were treated by daily subcutaneous injection of 10 mg kg^−1^ d^−1^ ISO for 7 days. For 7 days, 10 mg kg^−1^ d^−1^, 20 mg kg^−1^ d^−1^ ZB, 30 mg kg^−1^ d^−1^ Met, or equal volumes of Saline was gavaged daily as indicated. Met was used as a positive control; (**b**) Representative images of heart size. Scale bar: 5 mm; (**c**) Quantitative analysis of HW/BW ratio (*n* = 6); (**d**) Quantitative analysis of HW/TL ratio (*n* = 6); (**e**) Representative 1× and 40× images and quantification of picrosirius-red-stained collagen in the heart at the 8th day after ISO treatment for 24 h (*n* = 6). Scale bar (**upper**): 2 mm, scale bar (**lower**): 50 μm; (**f**) The mRNA levels of CTGF in heart tissues (*n* = 6); (**g**) The mRNA levels of Fn in heart tissues (*n* = 6); (**h**) The mRNA levels of α-SMA in heart tissues (*n* = 6); (**i**) The mRNA levels of Col-I in heart tissues (*n* = 6); (**j**) The mRNA levels of Col-III in heart tissues (*n* = 6). Quantification of CTGF, Fn, α-SMA, Col-I, and Col-III were normalized to GAPDH. The data are shown as mean ± SEM (one-way ANOVA with Tukey’s post-hoc multiple comparison tests). ****, *p* < 0.0001 vs. Saline; ###, *p* < 0.001, ####, *p* < 0.0001 vs. ISO. ISO, isoproterenol; Met, metoprolol; ZB, Zanubrutinib; HW/BW, the ratio of heart weight to body weight; HW/TL, the ratio of heart weight to tibia length; CTGF, Connective tissue growth factor; Fn, Fibronectin; Col-I, Collagen-I; Col-III, Collagen-III.

**Figure 3 molecules-28-06035-f003:**
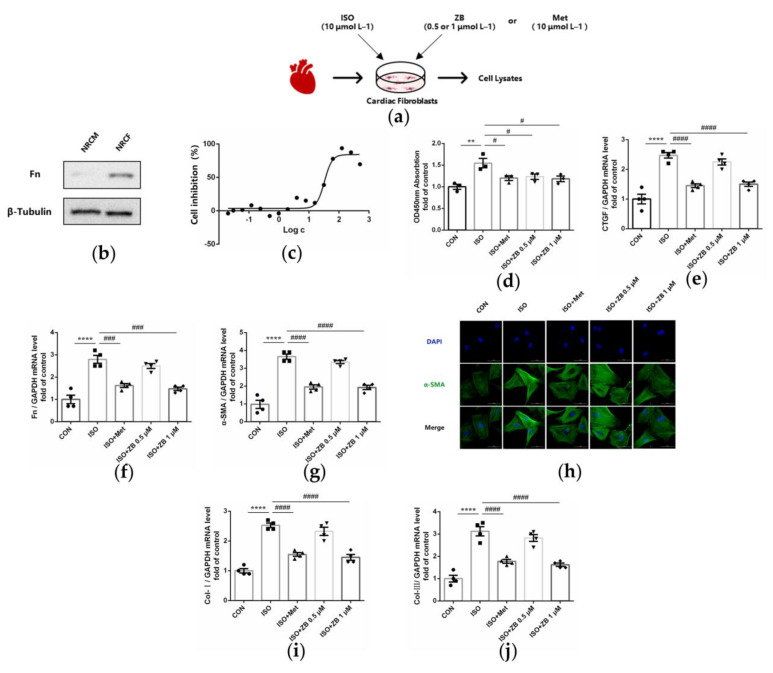
ZB inhibited ISO-induced cardiac fibroblast activation. (**a**) Dosing regimen in ISO-induced NRCFs. NRCFs were starved for 24 h, incubated with ZB (0.5 μmol L^−1^ or 1 μmol L^−1^), Met (10 μmol L^−1^), or equal volumes of DMSO for 1 h and then treated with ISO (10 μmol L^−1^) for 24 h. Met was used as a positive control; (**b**) Western Blot analysis of expression of Fn to ensure the purity of NRCFs; (**c**) CCK-8 analysis cell toxicity of ZB in NRCFs (*n* = 3); (**d**) CCK-8 analysis cell viability of ZB in NRCFs (*n* = 4); (**e**) The mRNA levels of CTGF in NRCFs (*n* = 4); (**f**) The mRNA levels of Fn in NRCFs (*n* = 4); (**g**) The mRNA levels of α-SMA in NRCFs (*n* = 4); (**h**) Immunofluorescence staining of α-SMA in NRCFs. Scale bar: 50 μm; (**i**) The mRNA levels of Col-I in NRCFs (*n* = 4); (**j**) The mRNA levels of Col-III in NRCFs (*n* = 4). Quantification of CTGF, Fn, α-SMA, Col-I, and Col-III were normalized to GAPDH. The data are shown as mean ± SEM (one-way ANOVA with Tukey’s post-hoc multiple comparison tests). **, *p* < 0.01, ****, *p* < 0.0001 vs. CON; #, *p* < 0.05, ###, *p* < 0.001, ####, *p* < 0.0001 vs. ISO. NRCFs, Primary neonatal rat cardiac fibroblasts; CON, control; ISO, isoproterenol; Met, metoprolol; ZB, Zanubrutinib; CTGF, Connective tissue growth factor; Fn, Fibronectin; Col-I, Collagen-I; Col-III, Collagen-III.

**Figure 4 molecules-28-06035-f004:**
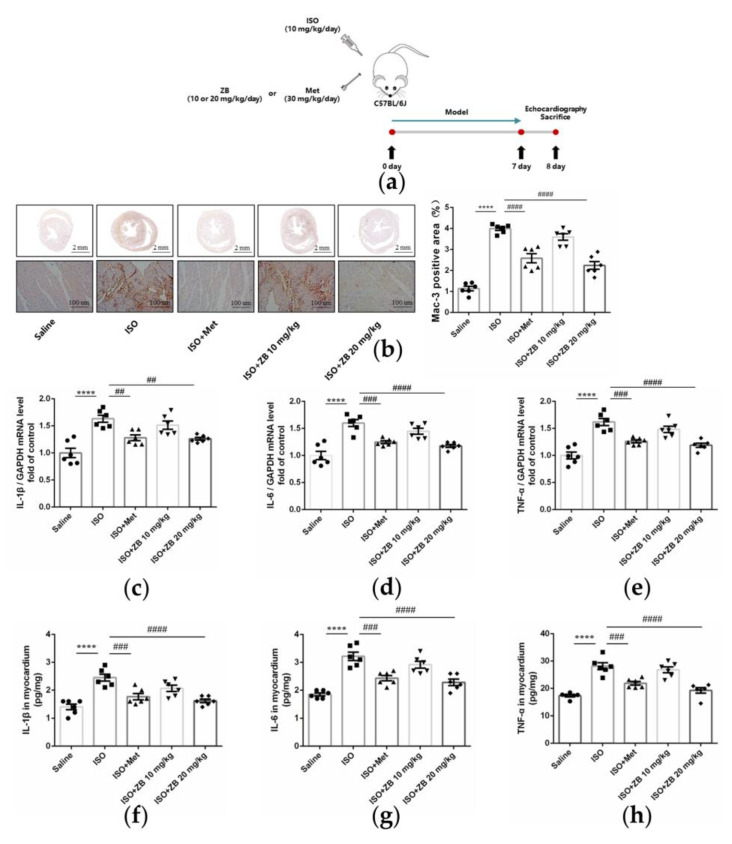
ZB attenuated ISO-induced cardiac inflammation. (**a**) Dosing regimen in ISO-induced cardiac inflammation model. Mice were treated by daily subcutaneous injection of 10 mg kg^−1^ d^−1^ ISO for 7 days. For 7 days, 10 mg kg^−1^ d^−1^, 20 mg kg^−1^ d^−1^ ZB, 30 mg kg^−1^ d^−1^ Met, or equal volumes of Saline was gavaged daily as indicated. Met was used as a positive control; (**b**) Representative 1× and 20× images of Mac-3 staining of heart sections. Scale bar (**upper**): 2 mm, scale bar (**lower**): 100 μm; (**c**) The mRNA levels of IL-1β in heart tissue (*n* = 6); (**d**) The mRNA levels of IL-6 in heart tissue (*n* = 6); (**e**) The mRNA levels of TNF-α in heart tissue (*n* = 6). Quantification of IL-1β, IL-6, and TNF-α were normalized to GAPDH; (**f**) The determination of IL-1β in heart tissue (*n* = 6); (**g**) The determination of IL-6 in heart tissue (*n* = 6); (**h**) The determination of TNF-α in heart tissue (*n* = 6). The data are shown as mean ± SEM (one-way ANOVA with Tukey’s post-hoc multiple comparison tests). ****, *p* < 0.0001 vs. Saline; ##, *p* < 0.01, ###, *p* < 0.001, ####, *p* < 0.0001 vs. ISO. ISO, isoproterenol; Met, metoprolol; ZB, Zanubrutinib; IL-1β, Interleukin-1β; IL-6, Interleukin-6; TNF-α, Tumor necrosis factor.

**Figure 5 molecules-28-06035-f005:**
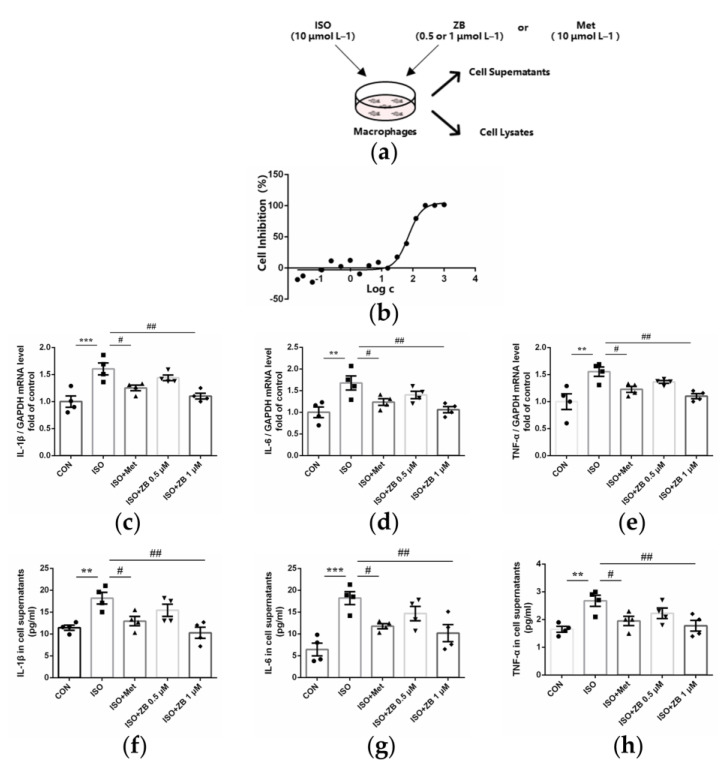
ZB reduced ISO-induced pro-inflammatory production. (**a**) Dosing regimen in ISO-induced RAW264.7 cells. RAW264.7 cells were starved for 12 h, incubated with ZB (0.5 μmol L^−1^ or 1 μmol L^−1^), Met (10 μmol L^−1^), or equal volumes of DMSO for 1 h and then treated with ISO (10 μmol L^−1^) for 24 h (*n* = 4). Met was used as a positive control; (**b**) Analysis of cell toxicity of ZB in RAW264.7 by CCK-8 (*n* = 3); (**c**) The mRNA levels of IL-1β in RAW264.7 cells (*n* = 4); (**d**) The mRNA levels of IL-6 in RAW264.7 cells (*n* = 4); (**e**) The mRNA levels of TNF-α in RAW264.7 cells (*n* = 4). Quantification of IL-1β, IL-6, and TNF-α were normalized to GAPDH; (**f**) The determination of IL-1β in RAW264.7 cell supernatants (*n* = 4); (**g**) The determination of IL-6 in RAW264.7 cell supernatants (*n* = 4); (**h**) The determination of TNF-α in RAW264.7 cell supernatants (*n* = 4). The data are shown as mean ± SEM (one-way ANOVA with Tukey’s post-hoc multiple comparison tests). **, *p* < 0.01, ***, *p* < 0.001 vs. CON; #, *p* < 0.05, ##, *p* < 0.01 vs. ISO. RAW264.7, mouse leukemia cells of monocyte macrophage; CON, control; ISO, isoproterenol; Met, metoprolol; ZB, Zanubrutinib; IL-1β, Interleukin-1β; IL-6, Interleukin-6; TNF-α, Tumor necrosis factor.

**Figure 6 molecules-28-06035-f006:**
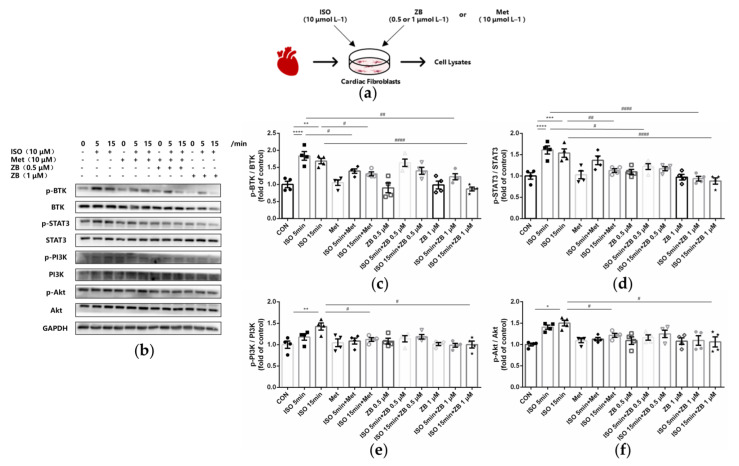
ZB inhibited ISO-induced cardiac fibroblasts activated by STAT3 and PI3K/Akt signaling pathways. (**a**) Dosing regimen in ISO-induced NRCFs. NRCFs were starved for 24 h, incubated with ZB (0.5 μmol L^−1^ or 1 μmol L^−1^), Met (10 μmol L^−1^), or equal volumes of DMSO for 1 h and then treated with ISO (10 μmol L^−1^) for 5 min or 15 min. Met was used as a positive control; (**b**) Western Blot analysis of BTK, STAT3, and PI3K/Akt signal pathways in NRCFs; (**c**) Quantification of BTK signal pathways in NRCFs (*n* = 4); (**d**) Quantification of STAT3 signal pathways in NRCFs (*n* = 4); (**e**) Quantification of PI3K signal pathways in NRCFs (*n* = 4); (**f**) Quantification of Akt signal pathways in NRCFs (*n* = 4). BTK, STAT3, PI3K, and Akt signal pathways were normalized to GAPDH. The ratio of phosphorylated protein to total protein reflects the activation of signaling pathways. The data are shown as mean ± SEM (one-way ANOVA with Tukey’s post-hoc multiple comparison tests). *, *p* < 0.05, **, *p* < 0.01, ***, *p* < 0.001, ****, *p* < 0.0001 vs. CON; #, *p* < 0.05, ##, *p* < 0.01, ####, *p* < 0.0001 vs. ISO. NRCFs, Primary neonatal rat cardiac fibroblasts; CON, control; ISO, isoproterenol; Met, metoprolol; ZB, Zanubrutinib.

**Figure 7 molecules-28-06035-f007:**
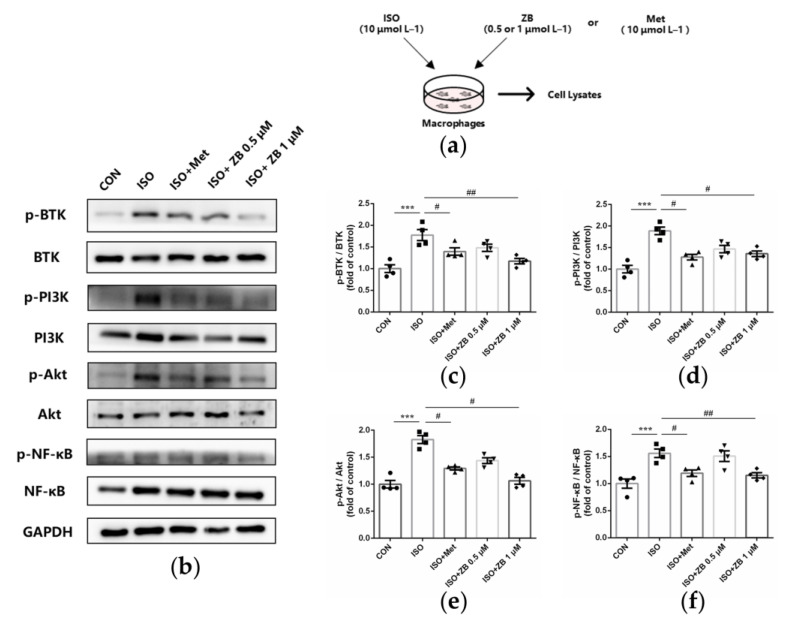
ZB inhibited ISO-induced macrophage pro-inflammatory production by PI3K/Akt and NF-κB signaling pathways. (**a**) Dosing regimen in ISO-induced RAW264.7 cells. RAW264.7 cells were starved for 12 h, incubated with ZB (0.5 μmol L^−1^ or 1 μmol L^−1^), Met (10 μmol L^−1^), or equal volumes of DMSO for 1 h and then treated with ISO (10 μmol L^−1^) for 24 h (*n* = 4). Met was used as a positive control; (**b**) Western Blot analysis of BTK, PI3K/Akt, and NF-κB signal pathways in RAW264.7 cells; (**c**) Quantification of BTK signal pathways in RAW264.7 cells (*n* = 4); (**d**) Quantification of PI3K signal pathways in RAW264.7 cells (*n* = 4); (**e**) Quantification of Akt signal pathways in RAW264.7 cells (*n* = 4); (**f**) Quantification of NF-κB signal pathways in RAW264.7 cells (*n* = 4). BTK, PI3K, Akt, and NF-κB signal pathways were normalized to GAPDH. The ratio of phosphorylated protein to total protein reflects the activation of signaling pathways. The data are shown as mean ± SEM (one-way ANOVA with Tukey’s post-hoc multiple comparison tests). ***, *p* < 0.001 vs. CON; #, *p* < 0.05, ##, *p <* 0.01 vs. ISO. RAW264.7, mouse leukemia cells of monocyte macrophage; CON, control; ISO, isoproterenol; Met, metoprolol; ZB, Zanubrutinib.

**Figure 8 molecules-28-06035-f008:**
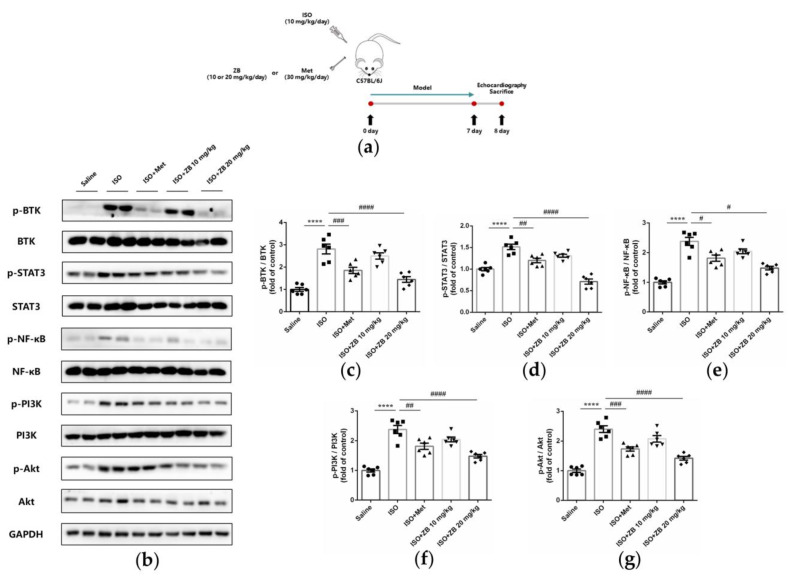
ZB inhibited ISO-induced cardiac fibrosis and inflammation by STAT3, PI3K/Akt, and NF-κB signaling pathways. (**a**) Dosing regimen in ISO-induced cardiac fibrosis and inflammation model. Mice were treated by daily subcutaneous injection of 10 mg kg^−1^ d^−1^ ISO for 7 days. For 7 days, 10 mg kg^−1^ d^−1^, 20 mg kg^−1^ d^−1^ ZB, 30 mg kg^−1^ d^−1^ Met, or equal volumes of Saline was gavaged daily as indicated. Met was used as a positive control; (**b**) Western Blot analysis of BTK, STAT3, NF-κB, and PI3K/Akt signal pathways in mice heart tissues; (**c**) Quantification of BTK signal pathways in heart tissues (*n* = 6); (**d**) Quantification of STAT3 signal pathways in heart tissues (*n* = 6); (**e**) Quantification of NF-κB signal pathways in heart tissues (*n* = 6); (**f**) Quantification of PI3K signal pathways in heart tissues (*n* = 6); (**g**) Quantification of Akt signal pathways in heart tissues (*n* = 6). BTK, STAT3, NF-κB, PI3K, and Akt signal pathways were normalized to GAPDH. The data are shown as mean ± SEM (one-way ANOVA with Tukey’s post-hoc multiple comparison tests). ****, *p* < 0.0001 vs. Saline; #, *p* < 0.05, ##, *p <* 0.01, ###, *p* < 0.001, ####, *p* < 0.0001 vs. ISO. ISO, isoproterenol; Met, metoprolol; ZB, Zanubrutinib.

**Figure 9 molecules-28-06035-f009:**
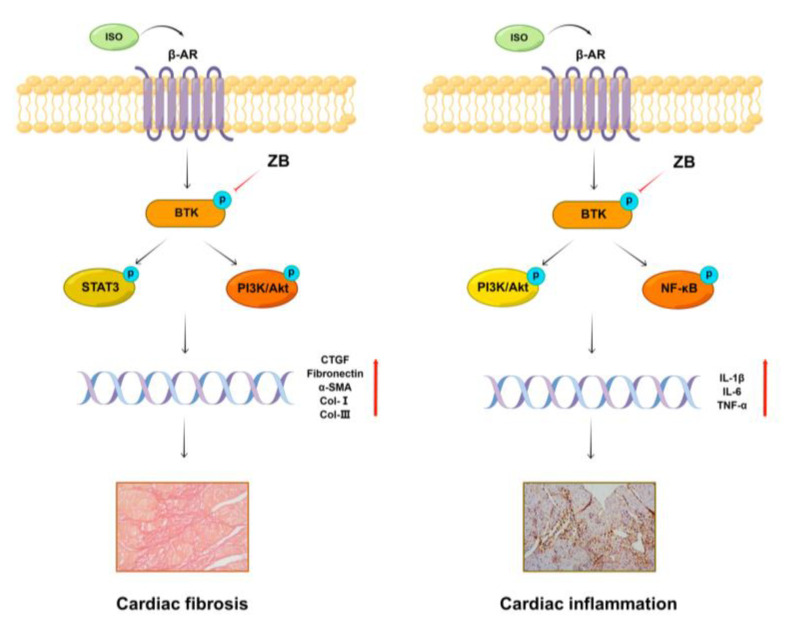
Working model of the effects of ZB on β-AR stimulation-induced cardiac fibrosis and inflammation.

**Table 1 molecules-28-06035-t001:** The list of gene primers.

Gene	Forward Primer Sequence (5′-3′)	Reverse Primer Sequence (3′-5′)
Mouse ANP	CTTCCAGGCCATATTGGAG	GGGGGCATGACCTCATCTT
Mouse BNP	ACAAGATAGACCGGATCGGA	AGCCAGGAGGTCTTCCTACA
Mouse CTGF	CCAACTATGATTAGAGCCAACTG	AGGCACAGGTCTTGATGAAC
Mouse Fibronectin	TCGGATACTTCAGCGTCAGGA	TCGGATACTTCAGCGTCAGGA
Mouse α-SMA	GTCCCAGACATCAGGGAGTAA	GTCCCAGACATCAGGGAGTAA
Mouse Collagen-I	ATGTGGACCCCTCCTGATAGT	ATGTGGACCCCTCCTGATAGT
Mouse Collagen-III	TGGTCCTCAGGGTGTAAAGG	GTCCAGCATCACCTTTTGGT
Mouse IL-1β	GAAATGCCACCTTTTGACAGTG	TGGATGCTCTCATCAGGACAG
Mouse IL-6	CTGCAAGAGACTTCCATCCAG	AGTGGTATAGACAGGTCTGTTGG
Mouse TNF-α	CAGGCGGTGCCTATGTCTC	CGATCACCCCGAAGTTCAGTAG
Mouse GAPDH	AGGTCGGTGTGAACGGATTTG	GGGGTCGTTGATGGCAACA

**Table 2 molecules-28-06035-t002:** The list of primary antibodies.

Antibody	Company	Item No.
p-BTK	Affinity	AF0841
BTK	Affinity	DF6472
p-NF-κB	Affinity	AF2006
NF-κB	Affinity	AF0874
p-PI3K	Affinity	AF3241
PI3K	Affinity	AF6241
p-Akt	Affinity	AF0016
Akt	Affinity	AF6261
p-STAT3	Affinity	AF0016
STAT3	Affinity	AF6294
Fibronectin	Proteintech	15613-1-AP
α-SMA	Affinity	AF1032
Collagen-I	Cell Signaling Technology	72026
Mac-3	BD Biosciences	550292
β-Tubulin	Affinity	AF7011
GAPDH	Affinity	AF7021

## Data Availability

The data presented in this study are available on request from the corresponding author. The data are not publicly available due to some reasons.

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
