# Peer review of "Zanubrutinib Ameliorates Cardiac Fibrosis and Inflammation Induced by Chronic Sympathetic Activation"

_molecules, 2023, doi:10.3390/molecules28166035_

Round 1

Reviewer 2 Report

The paper address the possible mode of action of zanubrutinib ameliorates cardiac fibrosis and inflammation using the isoproterenol model. As such the manuscript provides new information about possible mechanisms of action of zanubrutinib.

I have minor comments:

In the discussion the model, its limitations should be addressed. The level of catecholaminemia produced in the experimental setting used, is orders higher than in human pathology. Likewise, doses of the drugs, exceed those in clinical practice.

Figure 1 – bars are very small, can be seen/read comfortably at 300% magnification, so that the figure should be redrawn for better visibility. ISO, MET, ZB, mpk abbreviations should be expanded in the figure legend/subscript.

Figure 2 – bars are very small, can be seen/read comfortably at 200% magnification, so that the figure should be redrawn for better visibility. As above all abbreviations used should be expanded in the figure legend/subscript.

Figure 3 – bars are very small, can be seen/read comfortably at 200% magnification, so that the figure should be redrawn for better visibility. As above all abbreviations used should be expanded in the figure legend/subscript.

Figures from 4 to 8 – bars are very small, can be seen/read comfortably at 200% magnification, so that the figure should be redrawn for better visibility. As above all abbreviations used should be expanded in the figure legend/subscript.

Reviewer 3 Report

- Add more details for the reason of β-adrenergic activation in heart failure and upstream signaling. - How (TEC) can be tyrosine kinases 53 in hepatocellular carcinoma abbreviation? - Why did the authors prefer do picrosirius red instead of Masson trichrome? - Why data are presented by Mean and SEM instead of SD? - TGF-b is a main regulator in fibroblast activation. Why there is no any measurement of TGF in Fig.6? - Please clarify limitation of this study? - Can ZB can be used in other model of heart failure?

Round 2

Reviewer 3 Report

It could be accepted now.